# Bioactivity of Biomass and Crude Exopolysaccharides Obtained by Controlled Submerged Cultivation of Medicinal Mushroom *Trametes versicolor*

**DOI:** 10.3390/jof8070738

**Published:** 2022-07-17

**Authors:** Galena Angelova, Mariya Brazkova, Dasha Mihaylova, Anton Slavov, Nadejda Petkova, Denica Blazheva, Ivelina Deseva, Irina Gotova, Zhechko Dimitrov, Albert Krastanov

**Affiliations:** 1Department of Biotechnology, University of Food Technologies, 26 Maritsa Blvd., 4002 Plovdiv, Bulgaria; g_angelova@uft-plovdiv.bg (G.A.); dashamihaylova@yahoo.com (D.M.); a_krastanov@uft-plovdiv.bg (A.K.); 2Department of Organic and Inorganic Chemistry, University of Food Technologies, 26 Maritsa Blvd., 4002 Plovdiv, Bulgaria; antons@uni-plovdiv.net (A.S.); petkovanadejda@abv.bg (N.P.); 3Department of Microbiology, University of Food Technologies, 26 Maritsa Blvd., 4002 Plovdiv, Bulgaria; d_blazheva@uft-plovdiv.bg; 4Department of Analytical Chemistry and Physicochemistry, University of Food Technologies, 26 Maritsa Blvd., 4002 Plovdiv, Bulgaria; ivelina_hristova_vn@abv.bg; 5LB-Bulgaricum PLC, R&D Center, 1000 Sofia, Bulgaria; irina@gotov.bg (I.G.); jechkoelby@yahoo.com (Z.D.)

**Keywords:** *Basidiomycota*, *Trametes versicolor*, biomass, exopolysaccharide, bioactivity, prebiotic, antioxidant, anti-inflammatory

## Abstract

The aim of this study is to characterize the bioactivity of mycelial biomass and crude exopolysaccharides (EPS) produced by *Trametes versicolor* NBIMCC 8939 and to reveal its nutraceutical potential. The EPS (1.58 g/L) were isolated from a culture broth. The macrofungal biomass was rich in protein, insoluble dietary fibers and glucans. The amino acid composition of the biomass was analyzed and 18 amino acids were detected. Three mycelial biomass extracts were prepared and the highest total polyphenol content (16.11 ± 0.14 mg GAE/g DW) and the total flavonoid content (5.15 ± 0.03 mg QE/g DW) were found in the water extract. The results indicated that the obtained EPS were heteropolysaccharides with glucose as the main building monosaccharide and minor amounts of mannose, xylose, galactose, fucose and glucuronic acid. Fourier Transform Infrared Spectroscopy (FTIR) confirmed the complex structure of the crude EPS. Five probiotic lactic acid bacteria strains were used for the determination of the prebiotic effect of the crude EPS. The anti-inflammatory potential was tested in vitro using cell line HT-29. The significant decrease of IL-1 and IL-8 and increase of TGF-beta expression revealed anti-inflammatory potential of the crude exopolysaccharides from *T. versicolor*.

## 1. Introduction

The higher fungi of *Agaricomycetes* class (*Basidiomycota* division) are a huge source of a wide range of natural structurally diverse bioactive compounds with promising nutritious and therapeutic properties and have great potential as nutraceuticals [1,2] The basidiomycetes have a remarkable history and have been used in traditional Asian medicine for thousands of years because of their beneficial effects on human health, with no negative side effects [3,4]. Despite the fact that species from *Basidiomycota* were much less studied than *Ascomycota* representatives, approximately 700 basidiomycete species have been shown to have pharmacological activity due to the bioactive metabolites that they synthesize [5]. The main bioactive substances produced by basidiomycetes are homo- and hetero-polysaccharides, lectins, phenolic and flavonoid compounds, terpenoids, sterols and volatile organic compounds [2,6,7]. The diversity of bioactivities could be explained by the fact that mushrooms’ natural habitats are dark places and highly competitive environments, combined with constant attack by numerous bacterial species. Due to this, fungal strains create their own barrier systems and synthesize numerous bioactive metabolites that enable them to survive in their natural environment [8]. This is the reason why research related to the discovery of new components produced by higher fungi, as well as the determination of their in vitro and in vivo biological activity, has increased exponentially in the last ten years [4,9,10]. Beyond their nutritional value, mushroom metabolites have been shown to exhibit a plethora of bioactive properties such as antioxidant properties, anticarcinogenic activity, immunomodulatory action, prebiotic activity, antibacterial and antiviral activity, anti-inflammatory actions, antihypoglycemic activities, neuroprotective and anti-aging properties, etc. [2,6]. There is evidence that β-glucan polysaccharides, obtained from mushrooms, possess antiviral activity and could fight against coronavirus SARS-CoV-2 disease (COVID-19) due to different mechanisms of immunomodulation [11,12].

The mushroom species which are most often used as food for thousands of years and which demonstrate potential to synthesize bioactive components with therapeutic properties are *Ganoderma lucidum*, *Grifola frondosa*, *Lentinus edodes*, *Schizophyllum commune*, *Trametes versicolor*, *Hericium erinaceus*, *Inonotus obliquus*, *Phellinus linteus and Pleuortus ostreatus* [13,14,15,16,17]. Most of them are referred to as edible and the basidiocarps of *G. lucidum*, *T. versicolor* and *I. obliquus* are referred to as non-edible medicinal mushrooms because of their bitter taste and toughness [18,19,20].

The widespread cultivation techniques for macromycetes are solid-state ones which lead to fruiting body formation, a very long and laborious process. The proper controlled submerged cultivation in an appropriate nutrient medium is a promising approach leading to the increased production of macrofungal biomass and bioactive substances with consistent quality, decreased production time, reduced downstream processing cost, decreased contamination risks and assured sustainability possibility [2,21,22,23,24]. Serious efforts are being made to increase the yield of bioactive metabolites by the optimization of the cultivation medium composition and conditions [23,25,26,27,28], together with exploring the genetic mechanisms of the biosynthetic pathways and their regulation [29].

One of the most studied medicinal mushrooms in the past decade is *T. versicolor*, commonly known as Turkey tail. The species belongs to *Basidiomycota* phylum, order *Polyporales*, family *Polyporaceae*. Due to its high lignin degrading enzyme activities, the application of *T. versicolor* in mycoremediation processes is well known [30]. On the other hand, *T. versicolor* has been used in traditional medicine for centuries and lately is a part of modern cancer treatment because of its diversity of bioactive compounds with an enormous variety of chemical structure and physiological activities [31]. The best investigated carriers of biological activity are the polysaccharopeptides (PSPs) obtained from submerged batch cultivation of *T. versicolor* [32,33]. They exhibited antitumor, hepatoprotective and analgesic activities [34,35,36,37]. Additionally, it was reported that these compounds demonstrate significant immunomodulatory activity and have promising anti-diabetic properties as α-glucosidase inhibitors [7,38]. It was reported that the protein-bound polysaccharides (PSPs) were nontoxic during prolonged use in the treatment of cancers by suppressing DNA/RNA synthesis and enhancing immune function [32,39,40]. In recent years, *T. versicolor* has attracted the attention of scientists not only as a source of pharmacologically active substances, but also as adjuvants in conventional chemo-or radiation therapy to reduce their side effects or to enhance their potency [41]. 

The aim of our study was to investigate the composition of mycelial biomass and crude EPS obtained by submerged cultivation of a local macrofungal *T. versicolor* NBIMCC 8939 strain, which had previously been studied for its application in mycoremediation processes. The evaluation of the antioxidant activity of different biomass extracts together with the prebiotic and in vitro anti-inflammatory activity of the crude EPS is a prerequisite to reveal the nutraceutical potential of the mycelium obtained by submerged cultivation and the synthesized extracellular exopolysaccharides, in addition to the established mycoremediation capacity of *T. versicolor* NBIMCC 8939.

## 2. Materials and Methods

### 2.1. Mushroom

The macrofungal strain *T. versicolor* NBIMCC 8939 is part of the microbial collection of the Department of Biotechnology, University of Food Technologies, Plovdiv, Bulgaria. The strain was previously isolated, molecularly identified and deposited in the National Bank for Industrial Microorganisms and Cell Cultures. The strain was maintained at 4 °C on Mushroom Complete medium (MCM) containing 20.0 g/L of glucose, 0.5 g/L of KH_2_PO_4_, 1.0 g/L of K_2_HPO_4_, 0.5 g/L of MgSO_4_ × H_2_O, 2.0 g/L of peptone, 2.0 g/L of yeast extract, 2.0 g/L of agar, pH 4.8–5.2, and was subcultured every 30 days onto fresh medium.

### 2.2. Submerged Cultivation of T. versicolor NBIMCC 8939

Previously optimized by Angelova et al. [42], liquid Yeast Malts Extract Medium (YM) with the following content: 40.32 g/L of glucose, 3.51 g/L of yeast extract, 3.51 g/L of peptone, 3.00 g/L of malt extract, 7.09 g/L of (NH_4_)_2_SO_4_, 0.5 g/L of KCl, 0.5 g/L of MgSO_4_ × 7H_2_O and 0.01 g/L of FeSO_4_ × 7H_2_O, pH 6.0, was used for submerged cultivation of *T. versicolor* for biomass and EPS production. The vegetative inoculum was prepared from a 7-day old culture of *T. versicolor* grown on MCM agar slants. Each Erlenmeyer flask, containing 100 mL nutrient medium, was inoculated with vegetative biomass from a single MCM-slant culture. The submerged cultivation was carried out at 28 °C on a rotary shaker at 220 rpm for 9 days. Then the mycelium biomass was separated by filtration, washed with distilled water and lyophilized. The biomass was presented as grams dry weight per liter (g DW/L). The culture liquid was used for EPS precipitation.

### 2.3. Isolation of Exopolysaccharides

EPS were isolated from the culture liquid by overnight freezing at −18 °C followed by thawing at room temperature. The precipitated EPS were recovered by centrifugation (6000× *g* rpm, 20 min, 5 °C). The recovered crude EPS were dried in a laboratory dryer (PolEco, Poznań, Poland) at 30 °C for 12 h and the mass in grams was determined. The total carbohydrate content of the crude EPS was determined using the phenol-sulfuric method [43] with glucose as a standard and the protein content was assessed spectrophotometrically [44].

### 2.4. Preparation of Mycelial Biomass Extracts

Distilled water, ethanol (80%, *v*/*v*) and methanol were used as extracting solvents for obtaining biomass extract. The lyophilized biomass (1.5 ± 0.05 g) was ground, precisely weighed and mixed with 30 mL distilled water, 80% ethanol or methanol, respectively, then left in a laboratory shaker at 25 °C for 24 h. After that, the extracts were separated from the mycelium biomass by centrifugation. The biomass was treated with an additional 15 mL solvent at the same conditions. After the second centrifugation, the extracts were combined and kept at −8 °C.

### 2.5. Characterization of Mycelium Biomass

#### 2.5.1. Dietary Fibers Content Analysis

The total (TDF), soluble (SDF) and insoluble (IDF) dietary fibers were determined with K-TDFR-100A (Megazyme Int., Dublin, Ireland), according to AOAC method 991.43 “Total, soluble and insoluble dietary fibers in foods” (First action 1991) and AACC method 32–07.01 “Determination of soluble, insoluble and total dietary fibers in foods and food products” (Final approval 10-16-91) [45]. All values for TDF, SDF and IDF were expressed as g/100 g of a DW biomass

#### 2.5.2. Glucans Content Analysis

Contents of total, α-and β-glucans were determined in the biomass and EPS using the Mushroom and Yeast β-glucan Assay Kit (Megazyme Int., Dublin, Ireland), following the instruction of the manufacturer. Briefly, to estimate the total glucans content in the samples, 2 mL ice-cold 12 M sulfuric acid was used for hydrolysis of the polysaccharides in the samples for 2 h at 100 °C. After neutralization, hydrolysis proceeded to glucose using a mixture of exo-β-(1,3)-D-glucanase plus β-glucosidase in sodium acetate buffer (pH 4.5) for 1 h at 40 °C. Enzymatic hydrolysis with amyloglucosidase and invertase was conducted for α-glucan content estimation. To estimate total glucan and α-glucan content, glucose oxidase/peroxidase reagent was added and the absorbance of the samples was measured at 510 nm. The β-glucan content was calculated by subtracting the α-glucan from the total glucan content. All values of total, α- and β-glucans in biomass were expressed as g/100 g of a DW biomass.

#### 2.5.3. Total Polyphenol (TPC) and Flavonoid Content Analysis

The total polyphenol content of the fungal biomass extracts was analyzed using the Folin-Ciocalteu method of Kujala et al. [46] with some modifications. Each sample (1 mL) was mixed with 0.5 mL of Folin-Ciocalteu’s phenol reagent and 0.4 mL of 7.5% Na_2_CO_3_. The mixture was vortexed well and left for 5 min at 50 °C. After incubation, the absorbance was measured at 765 nm. The TPC in the extracts was expressed as mg gallic acid equivalent (GAE) per g dry weight (mg GAE/g DW).

The total flavonoid content of the fungal biomass extracts was evaluated according to the method described by Kivrak et al. [47]. An aliquot of 0.5 mL of the sample was added to 0.1 mL of 10% Al(NO_3_)_3_, 0.1 mL of 1 M CH_3_COOK and 3.8 mL ethanol. After incubation for 40 min at room temperature, the absorbance was measured (415 nm). Quercetin (QE) was used as a standard and the results were expressed as mg QE/g DW.

#### 2.5.4. Total Protein and Amino Acid Determination

The total nitrogen of the *T. versicolor* biomass was determined by the Kjeldahl method [45]. The total protein content was calculated by multiplying the total nitrogen by 4.38. The result was expressed as a percentage. 

The amino acid composition was determined by the method described by Tumbarski et al. [48]. The lyophilized biomass was subjected to acid hydrolysis using 6N HCl for 24 h at 105 °C. An aliquot of the hydrolysate was derivatised using AccQ-Fluor reagent Kit (Waters). The derivative was separated on RP AccQ-Tag™ silica-bonded amino acid column C18, 3.9 mm × 150 mm (Waters, Etten-Leur, The Netherlands), and conditioned at 37 °C using an ELITE LaChrom HPLC system (VWR™ Hitachi, Tokyo, Japan). A sample of 20 μL was injected and the elution of the amino acids was performed by the following gradient system: eluent A, buffer WAT052890 (Waters, Etten-Leur, The Netherlands) and eluent B, 60% acetonitrile (Merck KGaA, Darmstadt, Germany) with a constant flow rate of 1.0 mL/min. The amino acids were detected using a diode array detector (DAD) at 254 nm. The amino acid peaks were then analyzed using EZChrom Elite™ software [49] and the amino acid content was calculated based on the amino acid standard calibration curve (amino acid standard H, Thermo Fisher Scientific, Waltham, MA, USA). The results were expressed as mg AA/g sample and as a percentage [50].

### 2.6. Chemical and Structural Characterization of the Crude Exopolysaccharides

#### 2.6.1. Analysis of Monosaccharide Composition 

The determination of individual neutral sugars, galacturonic acid and glucuronic acid was performed as follows: 10 mg crude EPS was hydrolyzed with 15 mL 2 M trifluoroacetic acid (Merck KGaA, Darmstadt, Germany) for 3 h at 120 °C. In order to remove the trifluoroacetic acid, the hydrolysate was evaporated to dryness under a vacuum and dissolved in 10 mL deionized water; this procedure was repeated three times. The residue from the last evaporation was dissolved in 1 mL deionized water. The quantities of galactose, rhamnose, fucose, galacturonic and glucuronic acid were determined by chromatographic system ELITE LaChrome (Hitachi High-Tech Corporation, Ibaraki, Japan) HPLC with a VWR Hitachi Chromaster 5450 with refractive index detector using an Aminex HPX-85H column. The samples and standards were eluted with 5 mM H_2_SO_4_ (Merck KGaA, Darmstadt, Germany) at an elution rate of 0.5 mL/min, column temperature of 50 °C and detector temperature of 35 °C. The amounts of xylose and mannose were determined separately with the same chromatographic system using a Sugar SP0810 (Shodex^®^) column. The samples and standards were eluted with ultrapure water at an elution rate of 1.0 mL/min, column temperature of 85 °C and detector temperature of 35 °C.

#### 2.6.2. Molecular Weight Measurement

The molecular weight of the crude EPS was determined by size exclusion chromatography using an ELITE LaChrome (Hitachi High-Tech Corporation, Ibaraki, Japan) HPLC system with a VWR Hitachi Chromaster 5450 with refractive index detector and an OHpak SB-806M (Shodex^®^) column. The samples and standards were eluted with 0.1M NaNO_3_ at an elution rate of 0.8 mL/min, column, temperature 30 °C and detector temperature 35 °C. The column was equilibrated with Shodex pullulan (Showa DENKO, Tokio, Japan) standards (2 mg/mL) with molecular weights of 0.62 × 10^4^, 1.00 × 10^4^, 2.17 × 10^4^, 4.88 × 10^4^, 11.3 × 10^4^, 20.0 × 10^4^, 36.6 × 10^4^, and 80.5 × 10^4^ Da.

#### 2.6.3. Infrared (IR) Spectra

The IR spectra of the EPS (2 mg) were collected on a Fourier transform infrared (FTIR) spectrophotometer VERTEX 70v (Bruker, Bremen, Germany) in KBr pellets. The spectra were recorded in the 4000–400 cm^−1^ range at 132 scans with a spectral resolution of 2 cm^−1^.

### 2.7. Bioactivity Assays

#### 2.7.1. Determination of the In Vitro Antioxidant Activity (AOA) of Biomass Extracts 

DPPH radical scavenging activity

The ability of the mycelial biomass extracts to donate an electron and scavenge DPPH radicals was determined by the slightly modified method of Brand-Williams et al. [51]. Freshly prepared 4 × 10^−4^ M methanol solution of DPPH was mixed with the mycelium biomass extract at a ratio of 2:0.5 (*v*/*v*). The light absorption was measured at 517 nm at room temperature after 30 min incubation. The DPPH radical scavenging activity was presented as a function of the concentration of Trolox-Trolox equivalent antioxidant capacity (TEAC), and was defined as the concentration of Trolox having equivalent AOA expressed as the μM Trolox per g DW (μM TE/g DW).

ABTS radical cation decolorization assay

The radical scavenging activity of the extracts against radical cation (ABTS•+) was estimated according to a previously reported procedure with some modifications [52]. The results were expressed as TEAC values (μM TE/g DW).

Ferric reducing antioxidant power assay (FRAP)

The FRAP assay was carried out according to the procedure of Benzie and Strain [53]. The FRAP reagent was prepared fresh daily and was warmed to 37 °C prior to use. The absorbance of the reaction mixture was recorded at 593 nm after incubation at 37 °C for 4 min. The results were expressed as μM TE/g DW.

Copper reduction assay (CUPRAC)

CUPRAC assay was performed according to the method of Apak et al. [54]. Amounts of 1 mL of CuCl_2_ solution (1.0 × 10^−2^ M), 1 mL of neocuproine methanolic solution (7.5 × 10^−3^ M) and 1 mL NH_4_OOCCH_3_ buffer solution (pH 7.0) were added to a test tube and mixed; 0.1 mL of the mycelium biomass extract followed by 1 mL of water were added (total volume of 4.1 mL) and mixed well. Absorbance against a reagent blank was measured at 450 nm after 30 min. Trolox was used as standard and total antioxidant capacity of fungal biomass extracts was determined as μM TE/g DW.

#### 2.7.2. Prebiotic Activity of Crude Exopolysaccharides 

The prebiotic activity of the EPS was determined using the following probiotic bacteria: *Lactobacillus gasseri* S20, *Lactiplantibacillus plantarum* A, *Limosilactobacillus reuteri* BG, *Lacticaseibacillus casei* Y and *Lactobacillus acidophilus* A2. The effect of EPS was investigated through the cultivation of the lactic acid bacteria in 96-deep-well plates containing 2 mL nutrient medium (in g/L: proteose peptone–10.0, meat extract–8.0, yeast extract–4.0, sodium acetate–5.0, triamonium citrate–2.0, magnesium sulfate–0.2, manganese sulfate–0.05 and dipotassium sulfate–2.0; Tween, 80–1.0 mL; pH 6.2) with 1.0 g/L EPS. MRS broth (Merck KGaA, Darmstadt, Germany) was used as a positive control and the above-mentioned medium without carbon source was used as negative control. Every well was inoculated with 1% inoculum with OD_600_ = 0.5 ÷ 0.6. The plates were incubated at 37 °C and the optical density was determined after 24 h.

#### 2.7.3. In Vitro Determination of Anti-Inflammatory Potential of the Crude Exopolysaccharides

The HT-29 cells were cultured to monolayer in DMEM (Dulbecco’s modified Eagle’s medium, Gibco, UK), supplemented with 10% fetal bovine serum at 37 °C and 5% CO_2_. In about 90% of the cell monolayer, cells were passaged by incubation with 0.25% trypsin and 10 mM EDTA solution for 10 min at 37 °C. To determine the anti-inflammatory effect, eukaryotic cells were cultured in 48-well plates at a concentration of 2 × 10^5^ cells/mL. The medium was changed every 2 days for a total of 14 days, supporting not only a monolayer formation (3–4 days), but also the maturation of cellular receptors. The resulting monolayer was washed twice with phosphate buffered saline (PBS) buffer. In each well, 500 µL DMEM containing 2 mg/mL crude EPS were added and incubated for 20 h at 37 °C. After that, the cell-free supernatant was used for determination of cytokine expression. IL-1β, IL-8, TGF-β were assessed using enzyme-linked immunosorbent assays according to the manufacturer’s instruction (Diaclone, Ann Arbor, MI, USA). 

### 2.8. Statistical Analysis

All the experiments were conducted in triplicate and the values were expressed as mean ± SD. Statistical significance was detected by analysis of variance (ANOVA, Tukey’s test; value of *p* < 0.05 indicated statistical difference.

## 3. Results and Discussion

*T. versicolor* belongs to the medicinal white-rot fungi (*Basidiomycota* phyla, *Polyporales* order, *Polyporaceae* family) and is most commonly found as a saprophyte on dead hardwood. A widespread and common species in Bulgaria, this fungus grows most often on deciduous trees. It is a well-known medicinal macromycete due to many bioactive components which could have an impact on maintaining human health [10]. In our previous research, the mycoremediation capacity of *T. versicolor* NBIMCC 8939 and its ability to degrade lignocellulose and various polycyclic aromatic hydrocarbons were revealed [55,56,57,58].

### 3.1. Submerged Cultivation of T. versicolor NBIMCC 8939 for Mycelium Biomass and Exopolysaccharides Production

In the present study, the submerged cultivation of *T. versicolor* was conducted in a previously optimized nutrient medium where the carbon source was glucose at a concentration of 40.32 g/L [42]. *T. versicolor* grew mainly in pellet form and the cultivation lasted eight days. Relatively high quantities of mycelium biomass (10.22 ± 0.28 g DW/L) and EPS (1.84 ± 0.15 g/L) were produced. The biomass and EPS yield strongly depended on the strain and cultivation techniques as well as the different precipitation methods used for EPS recovery. Submerged cultivation is a reliable alternative for obtaining macrofungal biomass and valuable metabolites Moreover, this cultivation technique allows biologically active extracellular polysaccharides to be generated in addition to biomass [2,24,59,60,61,62,63]. According to Elisashvili [59], the main functions of EPS are to support mushroom adhesion to the substrate, to prevent hyphal dehydration, to store excess nutrients and to facilitate lignin degradation. The most reported biomass yields ranged between 0.1 and 16.0 g/L after 14 days of cultivation of different basidiomycetes strains [60,64]. According to the study of Osińska-Jaroszuk et al. [65], macromycetes belonging to the *Ascomycota* and *Basidiomycota* division produced EPS from 0.12 to 42.24 g/L. In the present investigation, the recovery of EPS was performed without the use of an organic solvent. We established that the freezing of the culture filtrate at-18 °C for 48 h led to the formation of an insoluble polysaccharide gel which was easily separated by centrifugation. The same findings were only reported by Bolla et al. [25]. According to Jaros et al. [60], due to the quick crystallization of pure water the concentration of EPS in the remaining liquid fraction increases until the macromolecules precipitate upon exceeding a solubility threshold. This method for EPS precipitation could be considered an environmentally friendly one because it does not need to use an organic solvent. It could be assumed that the polysaccharides obtained without solvent precipitation are more pure because the proteins and salts present in the medium do not co-precipitate and the natural structure of the EPS remains protected. This method is to be optimized in further surveys and compared with classical ethanol precipitation procedures. 

### 3.2. Characterization of Mycelium Biomass and Biomass Extracts

The chemical composition of macrofungal biomass is strain specific and strongly dependent on the environment and cultivation techniques [10]. Bioactive endo- and exopolysaccharides together with proteins, polyphenols, minerals, vitamins, sterols and immune-enhancing enzyme activities are distinctive characteristics of the mycelium of the species *T. versicolor* [41,66,67]. The macrofungal biomass could be considered more beneficial in detoxification processes and oxidative stress prevention than pure macrofungal extracts [68].

#### 3.2.1. Dietary Fibers and Glucans

In the current study, the content of dietary fibers and glucans was determined in the mycelium biomass obtained by submerged cultivation of *T. versicolor* and the results are presented in Table 1. They show that the insoluble dietary fibers predominate over the soluble ones, which were estimated to be 36.21 ± 0.45 g/100 g DW and 1.99 ± 0.15 g/100 g DW, respectively. The total glucan content was established to be 30.39 ± 1.25%, constituted by β-glucans (22.34 ± 0.16%) and α-glucans (6.79 ± 0.18%). Total glucans represented 76, 78% of the value of TDF and the β- glucans content was 61.70% of IDF. Minor amounts of chitin could also contribute to the TDF in the mycelium biomass.

The various health effects of dietary fibers with macrofungal origin have attracted the focus of scientists [69,70]. Mushroom cell walls are composed of chitin (poly (1→4)-β-linked polymer of N-acetyl-glucosamine) and polysaccharides–α-and β-D-glucans and manannas. These components of the macrofungal cell wall are referred to as non-digestible polysaccharides resistant to human gastrointestinal enzymes and can be considered dietary fibers.

Glucans are the most common representative IDF found in the fruiting body or mycelium of macromycetes, and their content depends on the species, environment, maturity and cultivation techniques and usually ranges between 3.1% and 46.5%, while the level of water-soluble ones (SDF) is usually less than 10% DM [69,71]. Glucans play a structure-forming role in mushroom cell walls [72,73] and β-glucans compose up to 50% and α-glucans up to 10% by dry weight of cell walls. β-glucans (usually β-1,3–glucans with β-1,6 branches) are the most isolated glucans from mushrooms, with a huge amount of significant bioactivities [74,75,76,77]. Less frequently studied are α-1,3 glucans and mixed α/β-D-glucans [78]. It is interesting to note that α-1,3 glucans found in the cell walls of higher mushrooms were specific only to *Dikarya*, and they were absent in the species belonging to the lower fungal phyla [73]. Lately, the immunomodulatory, antitumoral, hypoglycemic and hypolipidemic properties of higher fungi have been related to α-1,3-glucans [79,80].

#### 3.2.2. Phenolic and Flavonoid Content

Phenolic and flavonoid compounds are secondary metabolites with various physiological activities, such as antiatherogenic, anti-inflammatory, antimicrobial, antithrombotic, cardioprotective and vasodilator activities. These effects are attributed to their antioxidant, free radical scavenging and metal ion chelating properties [81].

To evaluate the total phenolic content and the total flavonoid content of the mycelium biomass of *T. versicolor*, three different extracts (80% ethanol, methanol and water) were obtained. The TPC ranged from 2.90 ± 0.06 to 16.11 ± 0.14 mg GAE/g DW and the TFC varied in the range from 0.52 ± 0.01 to 5.15 ± 0.03 mg QE/g DW (Table 2). 

The highest TPC and TFC were established for the water extract and the lowest for the methanol ones. The results were consistent with those reported by Pop et al. [31], 15.40 ± 0.81 and 46.22 ± 0.89 mg GAE/100 g DW content of total phenols in methanol and water extract of *T. versicolor,* respectively. The same authors did not even detect the presence of TFC in the methanol extract, and in water extract the total flavonoids were estimated at 14.77 ± 0.55 mg GAE/100 g DW, which is in line with the present results. Water as a solvent proved to be a good choice in *T. versicolor* biomass treatment, and in addition, water is recognized as the most environmentally friendly solvent to be considered in green chemistry processes [82].

#### 3.2.3. Protein and Amino Acid Content

The total protein content found in the biomass of *T. versicolor* in our study was 34.25 ± 0.25%, which is in the upper limit of the reported ranges [83]. The crude protein content for basidiomycetes varies between 11% and 49% [24]. New sources of edible proteins are constantly being sought to meet the increasing protein demand of a continuously growing world population [84]. The easily digestible proteins of mushrooms are determined as the main carrier of the mushroom’s nutritional value because they could provide all essential amino acid requirements [68]. Apart from the nutritional value, there are reports about bioactive proteins and peptides demonstrating pharmacological potential [68,85,86,87]. The protein content of the wild fruiting body and mycelium biomass obtained through submerged cultivation usually varies. The cultivation techniques and substrate composition strongly affected the mushroom’s chemical composition including protein content [2]. 

Amino acids were qualitatively and quantitatively analyzed and all essential amino acids were detected (Table 3). Their content represented 46.59% of all the amino acids present in the mycelial biomass. 

Miletić et al. [88] reported prevailing content of the amino acids L-lysine, L-glutamic, and L-aspartic acid. In our study it was serine and histidine content that dominated, accounting for 9.77 ± 0.04 and 10.89 ± 0.02 mg AA/g sample, respectively. The high protein content and the presence of eight essential amino acids could make the *T. versicolor* NBIMCC 8939 biomass a reliable vegan protein source alternative to animal protein. Compared to soy [89], it has slightly higher content of histidine, methionine and phenylalanine essential amino acids and similar content of threonine and isoleucine. However, lysine and leucine seem to be in greater deficiency in our strain biomass. Nevertheless, according to these results, the studied strain’s mycelial biomass can be considered as a potential alternative to animal protein and could be used as an ingredient for the fortification of various foods. Further analysis for protein digestibility and the presence of allergenic proteins is needed. Determination of immune-enhancing enzyme activity in the macrofungal biomass will also be of great importance for future studies.

### 3.3. Basic Chemical and Structural Characterization of the Obtained Crude Exopolysaccharides

Extracellular macrofungal polysaccharides are usually glucan type heteropolysaccharides and their bioactivity depends on structural and physicochemical characteristics as well as precipitation methods [2]. 

#### 3.3.1. Total Carbohydrate and Protein Content

In this study, the previously dried crude EPS were dissolved in distilled water to determine total carbohydrate and protein content. It was observed that the EPS were only partially soluble–around 45% was dissolved. The polysaccharide solubility could be negatively affected by precipitation or drying methods, and its decrease was probably consequent to conformational changes [60].

The soluble portion of the EPS consisted of 69.00 ± 0.52% total carbohydrates and 5.51 ± 0.75 g/100 g DW protein (Table 4). The insoluble part was hydrolyzed with 2 M trifluoroacetic acid for 2 h at 120 °C. The hydrolysate was evaporated to dryness; 20 mL deionized water was added, left for 1 h at room temperature and again evaporated to dryness (the procedure was repeated three times). After the last evaporation, the dry mass was dissolved with 1 mL water and the protein content was determined. The results suggested that the insoluble part of the crude EPS contained 8.24 ± 1.05 g/100 g DW protein. The soluble part of the EPS was used for further analyzes, molecular weight determination and bioactivity analysis. The insoluble part will be the subject of future investigations. 

#### 3.3.2. Monosaccharide Content

The pharmacological activity of fungal EPS is strongly affected by monosaccharide content, molecular weight, solubility, viscosity and advanced structure. [2,90]. The most extensively reported monosaccharide compositions involved glucose, mannose, galactose, xylose, arabinose, rhamnose, and fucose. Less frequently found components of macrofungal polysaccharides were fructose, ribose, glucuronic acid, galacturonic acid, N-acetyl-glucosamine and N-acetyl-galactosamine, but it should be noted that the composition may vary according to mushrooms’ special and culture conditions [90].

The main building monosaccharide of the crude EPS obtained in the present study was glucose, which indicated that it was a glucan type polysaccharide. The minor amounts of mannose, xylose, galactose, fucose and glucuronic acid found suggested that the EPS were heteropolysaccharides (Table 4).

Other authors also reported that glucose was the dominant sugar in the monosaccharide profile of exo- and endopolysaccharides synthesized by *T. versicolor* [66,67,91,92], while Wang et al. [93] reported only glucose content in extracted EPS. The most reported carbohydrates, other than glucose, included mannose, xylose, galactose, rhamnose, arabinose and fucose [94,95]. The presence of mannose and fucose is an important sign for bioactivity because according to some findings, mannose-rich EPS could stimulate the immune system through macrophage located receptors [90,96]. According to Xiao et al. [97], L-fucose plays a significant role in the anticancer and anti-inflammatory activity of bacterial EPS.

#### 3.3.3. Molecular Weight

The soluble fraction of crude EPS was initially dissolved in 0.1M NaNO_3_ (which was used as the mobile phase during gel permeation chromatography) at 4 mg/mL for molecular weight determination. This fraction was heterogeneous with three major peaks found on the chromatogram: one with the highest molecular mass: 2.45 × 10^4^ Da (26.02% from total peak areas); the second one (having 39.57% from total peak areas) with 5.3 × 10^3^ Da molecular mass and the third one (33.98%) below 1 kDa (0.3 × 10^3^). The low molecular fraction is most probably constituted by proteins and/or partially degraded polysaccharide moieties.

It is interesting to note the observation by Wang, Chen and Hua [98] that the bioactivity of mushroom polysaccharides can be classified into three groups: (a) antidiabetic activity, when the molecular mass is between 3 and 5 kDa; (b) anti-inflammatory activity, when the molecular mass is between 10 and 100 kDa; (c) antitumor activity, when the molecular mass is over 30 kDa. Based on these conclusions and our data for the molecular mass of the EPS, we could expect that the crude exopolysaccharide isolated from *T. versicolor* might possess anti-inflammatory properties, which were later confirmed.

#### 3.3.4. FTIR Spectra of Exopolysaccharides

The bands typical of carbohydrates were found in the EPS isolated from *T. versicolor.* The FTIR spectrum is shown in Figure 1 and the assignment of characteristic bands were summarized in Table 5. Duvnjak et al. [99] reported bands close to our bands for β-bonds in the polysaccharide structure and protein presence for a batch culture of *T. versicolor*. The absorption band at 3329 cm^−1^ indicated the presence of the hydroxyl group (–OH) characteristic of molecular interactions of polysaccharide chains. The two bands towards 1654 and 1437 cm^−1^ were assigned to the deprotonated carboxylic group (–COO^−^). The presence of a band at 1741 cm^−1^ could be assigned to COO– from uronic acid or a delta-lactone structure. 

The intense bands at 2931 cm^−1^ were due to the C–H stretching vibrations. Two bands at 1654 and 1540 cm^−1^ were observed and were assigned to amide I and amide II vibrations of proteins. Similar observations were reported by Synytsya et al. [100] for glucans from fruiting bodies of cultivated mushrooms *Pleurotus ostreatus* and *Pleurotus eryngii*. The high absorbency ranged from 1097 cm^−1^. The IR bands in the region of 1000–1200 cm^−1^ were connected with stretching vibrations of C–O–C and C–O from the pyranose ring (Table 5, Figure 1). The bands at 1026, 1132, and 1186 cm^−1^ suggested the presence of C–O from pyranose ring, and the band at 1073 cm^−1^ was characteristic of the presence of β-glucans. In our case, the band at 1097 cm^−1^ was typical for the C–O–C and C–O–H vibrations reported previously by Du et al. 2017.

There were clearly visible bands at 935, 890, 870, and 816 cm^−1^ in the fingerprint region of our EPS spectrum from *T. versicolor* that revealed the presence of α- and β-glucans (Table 5 and Figure 1). The presence of α-glycosidic bonds was detected in the region 920–930 cm^−1^ for *T. versicolor* and *L. edodes* polysaccharides extract [101]. According to Baeva et al. [102], the band at 890 cm^−1^ was attributed to (1→3)-β-D-glucans, while the bands at 850 and 929 cm^−1^ were characteristic of α-D-glucans, and the band at 822 cm^−1^ specifically to (1→3)-α-D-glucan. Synytsya et al. [100] reported the bands of α-1,3-glucan at 930, 850, 822, 542, 448 and 421 cm^−1^**.**

In addition, bands at 870 and 816 cm^–1^ in the spectra confirmed the CH_2_ ring vibration of the β-anomer. C–H out of plane deformation, presence of gluco- and galacto-configuration, and two skeletal vibration bands at 543 and 454 cm^−1^ were also attributed to the latter α-D-glucan. The bands at 1654, 1418 and 935 cm^−1^ in our spectrum were near to the observation for 1657, 1417 and 940 cm^−1^ arising from chitin’s polysaccharide structure [100]. Therefore, the complex and overlapping bands in exopolysacchirides from the studied *T. versicolor* revealed the complex structure of this crude EPS, including α- and β-bonds typical of glucans and residues from chitin and binded proteins. In addition, the bands at 645 and 438 cm^−1^ revealed the presence of α- and β-xylose, which was also detected in the monosaccharide composition of EPS from *T. versicolor* NBIMCC 8939.

### 3.4. Bioactivity of Mycelium Biomass and Crude Exopolysaccharides from T. versicolor NBIMCC 8939

#### 3.4.1. In Vitro Antioxidant Activity (AOA) of Mycelium Biomass Extracts

In the present study, the antioxidant activity assessed by four in vitro assays varied between the methods (Table 6). According to Elmastas et al. [103], the antioxidant activity of different mushroom species against various antioxidant systems in vitro could be related to the phenolic compounds. In line with this and the aforementioned results for TPC and TFC, the highest antioxidant activity was estimated for the water extract and the lowest for the methanol ones. However, the results of the FRAP assay were contradictory. According to it, the most active extract was the methanol one, which reveals a curious circumstance and explains why the authors have always conducted several in vitro methods for antioxidant activity assessment. This is probably due to some flaws of FRAP: the assay is performed at acidic pH (3.6) to maintain iron solubility, and the FRAP assay does not measure thiol antioxidants, such as glutathione [104]. Thus, FRAP may not give comparable values to those under physiological conditions. On the other hand, the other three assays conducted to assess the antioxidant capacity revealed the higher potential of the water extract, which could probably be related to the higher phenolic and flavonoid content.

Given the established biological activity of the mycelial biomass of *T. versicolor*, it can be considered as a potential source of antioxidant activity.

In addition, the aqueous extract with the highest antioxidant capacity can be a good natural source of antioxidant(s) based on the well-known possible interaction of polyphenolic and polysaccharide compounds, which leads to the formation of water-soluble triterpenoids [105]. Further investigations are needed to elucidate the chemical origin of the compounds responsible for antioxidant activity.

#### 3.4.2. Prebiotic Activity of Exopolysaccharides

Polysaccharides from higher fungi have the potential for prebiotic activity because they usually cannot be digested in the human gastrointestinal tract. Some authors report the beneficial effect of polysaccharides from *G. lucidum*, *L. edodes* and *T. versicolor* on the gut probiotic microflora [2,69,106]. The first to provide clinical study evidence for the prebiotic properties of a *T. versicolor* polysaccharopeptide were Pallav et al. [107]. Since then, interest in the ability of macrofungal polysaccharides to modify the human intestinal microbiome has grown and the possibility of their stimulating beneficial microflora at the expense of harmful ones positions them as valuable ingredients for the production of functional foods and nutraceuticals [108].

All probiotic strains investigated in the present study were able to develop in the presence of EPS without the addition of other carbon sources. The highest optical density of 3.026 was detected for *Lactiplantibacillus plantarum* A (Figure 2), followed by *Lactobacillus gasseri* S20 (OD_600_ = 2.129). 

These strains exhibited significantly better growth when the fungal EPS were the only carbon source in the medium in comparison with a nutrient medium without a carbon source. *Lacticaseibacillus casei* Y and *Lactobacillus acidophilus* A2 demonstrated optical densities of 1.392 and 1.335, respectively, which were closer to those of their negative controls, but still a prebiotic effect was observed (Figure 2). For *Limosillactobacillus reuteri* BG, the growth in the presence of EPS and without carbon source was comparable and practically no probiotic activity was detected.

All strains exhibited their best development in MRS broth in comparison with the tested EPS concentrations. Nevertheless, for four of the studied lactic acid bacteria, the prebiotic effect of the EPS was demonstrated. The investigation of the ability of a larger number of probiotic species to utilize the studied EPS would give additional data on the spectrum of their prebiotic activity. Further clinical studies are also needed to definitively prove the effect of the polysaccharides from *T. versicolor* NBIMCC 8939 on the human intestinal microbiome. 

#### 3.4.3. In Vitro Anti-Inflammatory Potential of the Crude Exopolysaccharides

Inflammation is the protective response of the innate immune system to tissue injury or external factors [109]. Lately, the strong relationship between inflammation and cancer has been demonstrated [110]. The inflammatory process is mediated by cytokines called pro-inflammatory cytokines such as TNFα, IL-1, IL-6, and IL-8 [111]. The scope of several investigations was the determination of the anti-inflammatory potential of different macrofungi or their metabolites [112,113,114]. The most significant contributors to the anti-inflammatory activity of higher mushrooms tend to be endo- and exopolysaccharides, terpenoids and phenolic compounds [114]. It has been proposed that the potent anti-inflammatory activity of polysaccharides was possibly due to the inhibition of proinflammatory cytokines or enhancing of the production of anti-inflammatory cytokines [115].

In the present study, three particular cytokines (IL-1β, IL-8, TGF-β) were selected to investigate the in vitro anti-inflammatory activity of crude EPS according to two circumstances. Firstly, these three cytokines are involved in the signaling of inflammatory processes; secondly, the selected cytokines could be produced by the epithelial cell line HT-29. IL-1β and IL-8 are among the cytokines that mediate the inflammatory processes, and TGF-beta is one of the signal peptides associated with anti-inflammatory processes [116,117].

The results of the in vitro determination of anti-inflammatory activity of the crude exo-polysaccharides are presented in Table 7. 

The presence of 2 mg/mL crude EPS in DMEM led to a significant decrease in the concentration of the proinflammatory cytokines IL-8 and IL-1β, while at the same time a 2.5-fold increase in the concentration of TGF-β was observed. The increase of TGF-β could be associated with anti-inflammatory properties of the analyzed EPS. 

Mushroom EPS fractions with molecular weight between 10–100 kDa were mostly associated with anti-inflammatory properties [97]. The previously established M_w_ of the studied exopolysaccharide was 2.45 × 10^4^ Da, which was in accordance with the abovementioned findings. Similar results were reported by Bains et al. [10]. According to their research, a crude EPS obtained by T. versicolor was demonstrated to suppress the expression of pro-inflammatory cytokines such as IL-1β, IL-2, TNF α, and IFN-y). Contrary to our results, other studies found that compounds from mushrooms such as polysaccharide peptides provoke in vitro secretion of several proinflammatory cytokines such as IL-1, IL-2, IL-6, IL-8 and TNF-α [117]. Jin et al. [107] also reported about pro-inflammatory activity of metabolites secreted by the species *T. versicolor*. In this study the anti-inflammatory properties of EPS from *T. versicolor* NBIMCC 8939 were demonstrated by assessing the concentrations of the proinflammatory cytokines IL-1 and IL-8, which appeared to be decreased; at the same time, the concentration of TGF-β increased.

## 4. Conclusions

In the present study, a submerged cultivation of the medicinal fungus *T. versicolor* NBIMCC 8939 was carried out. The bioactivity potential of macrofungal biomass and crude EPS was revealed and the possibility for them to be used as dietary supplements with healthy effects, in addition to in mycoremediation applications, was demonstrated. Based on the paradigm that the health-promoting properties of food do not depend on just one component and usually are the result of the synergistic action of many bioactive compounds, no specific purification procedures were applied. The findings of this study give us the confidence to conclude that combining the macrofungal biomass and crude exo-cellular polysaccharide obtained by submerged cultivation of *T. versicolor* NBIMCC 8939 could represent an interesting approach to obtaining minimally treated natural dietary supplements with significant health effects.

## Figures and Tables

**Figure 1 jof-08-00738-f001:**
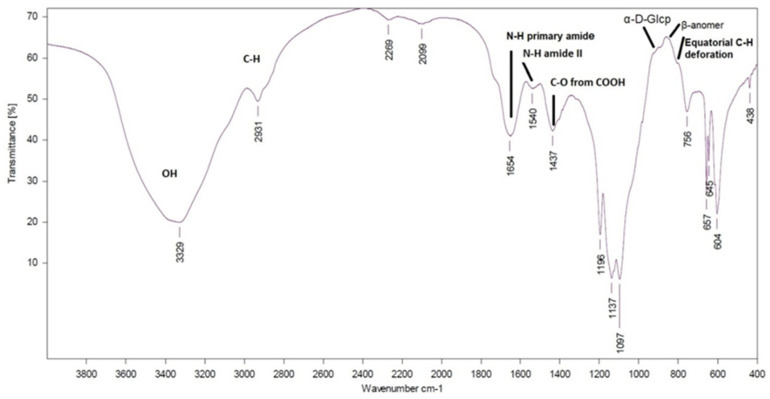
FTIR spectrum of exopolysacchirides (EPS) from *T. versicolor* NBIMCC 8939.

**Figure 2 jof-08-00738-f002:**
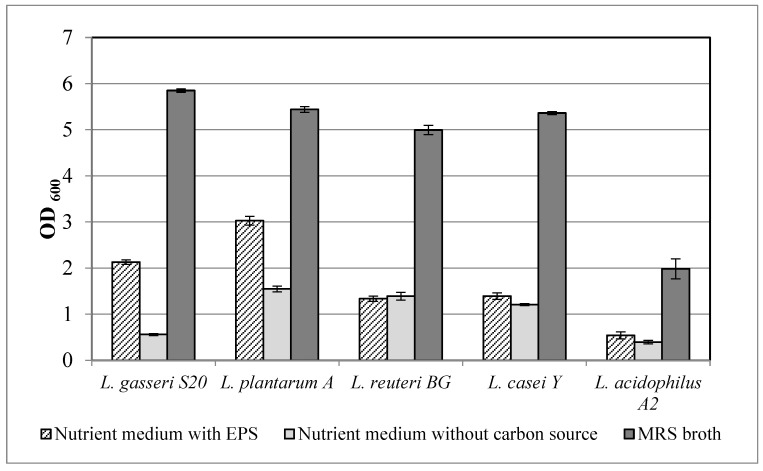
Utilization of exopolysaccharides (EPS) by probiotic lactic acid bacteria.

**Table 1 jof-08-00738-t001:** Dietary fiber and glucan content in the mycelium biomass of *T. versicolor* NBIMCC 8939.

*T. versicolor*	Dietary Fibers, g/100 g DW			Glucans, g/100 g DW		
	TDF	IDF	SDF	Total	α	β
Mycelium biomass	39.53 ± 0.61	36.21 ± 0.45	1.99 ± 0.15	30.39 ± 1.25	6.79 ± 0.18	22.34 ± 0.16

β-Glucan (% *w*/*w*) = Total glucan (% *w*/*w*) − α-glucan (% *w*/*w*).

**Table 2 jof-08-00738-t002:** Total phenolic content (TPC, mg GAE/g DW) and total flavonoid content (TFC, mg QE/g DW) of *T. versicolor* NBIMCC 8939 extracts.

Extract/Assay	TPC	TFC
80% ethanol	3.41 ± 0.02	1.26 ± 0.02
methanol	2.90 ± 0.06	0.52 ± 0.01
water	16.11 ± 0.14	5.15 ± 0.03

**Table 3 jof-08-00738-t003:** Amino acid composition of mycelial biomass of *T. versicolor* NBIMCC 8939.

Amino Acid	mg/g Sample	%
Asp (D)	7.09 ± 0.02	7.69
Ser (S)	9.77 ± 0.04	10.60
Glu (E)	6.31 ± 0.01	6.85
Gly (G)	3.08 ± 0.03	3.34
His * (H)	10.89 ± 0.02	11.82
Arg (R)	7.67 ± 0.01	8.32
Thr * (T)	7.03 ± 0.01	7.62
Ala (A)	6.49 ± 0.03	7.04
Pro (P)	3.55 ± 0.03	3.85
Cys (C)	0.03 ± 0.01	0.04
Tyr (Y)	5.23 ± 0.02	5.67
Val * (V)	4.66 ± 0/02	5.05
Met * (M)	1.59 ± 0.02	1.72
Lys * (K)	2.75 ± 0.01	2.98
Ile * (I)	6.12 ± 0.04	6.64
Leu * (L)	0.72 ± 0.02	0.78
Phe * (F)	9.20 ± 0.02	9.98

* Essential amino acids.

**Table 4 jof-08-00738-t004:** Monosaccharide composition and molecular weight of soluble crude exopolysaccharides (EPS) from *T. versicolor* NBIMCC 8939.

Parameters	Crude EPS
Total carbohydrate,%	69.00 ± 0.52
Protein, g/100 DW	5.51 ± 0.75
**Monosaccharide composition**, mg/g	
Glucose	38.85 ± 1.12
Mannose	4.24 ± 1.34
Xylose	3.86 ± 0.75
Galactose	3.54 ± 0.59
Fucose	2.37 ± 0.71
Glucuronic acid	2.10 ± 0.78
**Molecular weight**, Da	
Peaks 1 (26.02%)	2.45 × 10^4^ Da
Peaks 2 (39.57%)	0.53 × 10^4^ Da
Peaks 3 (33.98%)	0.30 × 10^3^ Da

The amount of sugars is given in mg/g EPS.

**Table 5 jof-08-00738-t005:** Assignment of characteristic bands in FTIR spectrum of exopolysacchrides (EPS) from *T. versicolor* NBIMCC 8939.

Bands, cm^−1^	Experimental Bands of EPS, cm^−1^	Assignment
3200–3400	3329	intermolecular H-brige between OH groups, free OH groups
2933–2981	2931	asymetric stretching vibrations, C–H (CH_2_)
2850–2904	2889	symetric stretching vibrations, C–H (CH_2_)
1745–1735	1741	C=O stretching vibration
1664–1634	1654	absorption of water in polymer, N–H deformation of amide I from protein of chitin
1541	1540	Amide II from protein or chitin
1455–1470	1437	symetric stretching vibrations, C–H (CH_2_) in pyranose ring, in-plane bending of o–н (OH), C–O from COO^−^
1416–1430	1418	CH_2_ scissor vibration
1382	1390	symmetric bending of CH_3_
1125–1162	1137	νC–O–C_as_ (C–O–C), stretching of glycosidic bonds
1015–1060	1097	stretching of C–O (C–O)
1107–1010	1107	Ring assimetric streching
985–996	986	stretching of C–O (C–O)
925–930	937	pyranose ring vibration; α-D-Glcp (glucose residue in polisacchride chain); scissoring vibrations in C–H in α-configuration
890	890	β-D-glucopyranose
870	870	Rocking of CH_2_ in the ring, β-anomer, galactopyranose derivatives
830–810	816	C–H equatorial deformation of anomer, β-bonds
		C–H out-of-plane deformation, gluco- and galacto- configuration of unit
770–730	756	C–H out-of-plane deformation, gluco- and galacto- configuration of unit

**Table 6 jof-08-00738-t006:** In vitro antioxidant activity of *T. versicolor* NBIMCC 8939 extracts according to DPPH, ABTS, FRAP and CUPRAC (µM TE/g DW).

Extract/ Assay	TEAC_DPPH_	TEAC_ABTS_	TEAC_FRAP_	TEAC_CUPRAC_
80% ethanol	2.21 ± 0.02	8.55 ± 0.13	4.16 ± 0.05	17.68 ± 0.07
methanol	2.37 ± 0.02	8.59 ± 0.31	6.12 ± 0.11	30.81 ± 1.06
water	5.63 ± 0.11	28.16 ± 0.49	4.49 ± 0.17	52.21 ± 0.28

**Table 7 jof-08-00738-t007:** The anti-inflammatory activity of crude exopolysaccharide (EPS) by *T. versicolor* NBIMCC 8939.

Sample	IL-1β, pg.mL^−1^	IL-8, pg.mL^−1^	TGF-β, pg.mL^−1^
2 mg/mL EPS in DMEM	7.8 ± 0.8	318 ± 34	5848 ± 41
DMEM (control)	14.8 ± 1.2	509 ± 42	2337 ± 17

## Data Availability

The raw/processed data required to reproduce these findings cannot be shared at this time as the data also form part of an ongoing study.

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
