# Peer review of "Bioactivity of Biomass and Crude Exopolysaccharides Obtained by Controlled Submerged Cultivation of Medicinal Mushroom Trametes versicolor"

_jof, 2022, doi:10.3390/jof8070738_

Round 1

Reviewer 1 Report

In my opinion the manuscript is very well organized and presents interesting and important issue. I recommend to accept this manuscript.

L619 "phenoloc" should be corrected

Author Response

We would like to thank Reviewer 1 for his/her positive review on our work. Regarding to the word „phenolic“ is no longer in the manuscript since the conclusion was revised.

Reviewer 2 Report

Lo stress ossidativo è stato ora identificato come la causa di malattie e cancro. Numerosi sono i lavori in letteratura che dimostrano le proprietà di funghi e sostanze vegetali che agiscono sulla riduzione dei ROS e come terapia e chemioterapica. Il manoscritto è molto interessante e dimostra le proprietà salutari degli alimenti, da cui derivano gli integratori. Il manoscritto è accettato in forma presente per la pubblicazione

Author Response

We would like to thank Reviewer 2 for the positive review.

Reviewer 3 Report

Dear Authors,

Reading the m/s, my general impression is very good. The subject of the present article is very interesting. The m/s is well organized; M & M are clearly presented, data are well analyzed and results are presented adequately. English language and style is quite good. However, more discussion is needed, where the results on T. versicolor will be compared to those of other mushroom species. Therefore, minor revision is my suggestion.

Author Response

We would like to thank Reviewer 3 for the positive review. The aim of our investigation was to study the bioactivity of the biomass and EPS obtained by this specific strain of Trametes versicolor, which was used until now for mycoremediation investigations. The group of medicinal fungi is extraordinarily diverse and they possess a vast spectrum of biological activities. Our purpose was to compare our results exclusively with those for other Trametes sp., and where applicable the data was compared to other mushroom species (for example section 3.3.4 and section 3.4.2.)

Reviewer 4 Report

1. A space is required between the unit and the number, such as 15mL on line 141.

2. In the Results and Discussion section, it is recommended to briefly describe your own research results before analyzing and discussing them.

3. In the Results and Discussion section, it is recommended to briefly describe your own research results before analyzing and discussing them. For example, lines 284-291, this is not the results of your research, it is recommended to delete it or put it in the analysis or discussion.

4. The word "diatery" in the conclusion part (line 614) is wrong, please correct it.

5. The conclusion is too long, it is recommended to be concise.

6. This manuscript requires linguistic polish.

Author Response

We would like to thank Reviewer 4 to for the remarks on our research. We hope that the added corrections will make the results and discussion section more informative to the readers. The remarks were taken into account when revising the manuscript and all errors were corrected.

Round 2

Reviewer 4 Report

This revised manuscript coule be accepted.